# On the fluorescence enhancement of arch neuronal optogenetic reporters

Leonardo Barneschi[1], Emanuele Marsili [1,2,7], Laura Pedraza-González [1,8], Daniele Padula [1], Luca De Vico [1], Danil Kaliakin [3], Alejandro Blanco-González [3], Nicolas Ferré[4], Miquel Huix-Rotllant [4], Michael Filatov [5] & Massimo Olivucci [1,3,6] ✉

The lack of a theory capable of connecting the amino acid sequence of a light-absorbing protein with its fluorescence brightness is hampering the development of tools for understanding neuronal communications. Here we demonstrate that a theory can be established by constructing quantum chemical models of a set of Archaerhodopsin reporters in their electronically excited state. We found that the experimentally observed increase in fluorescence quantum yield is proportional to the computed decrease in energy difference between the fluorescent state and a nearby photoisomerization channel leading to an exotic diradical of the protein chromophore. This finding will ultimately support the development of technologies for searching novel fluorescent rhodopsin variants and unveil electrostatic changes that make light emission brighter and brighter.

The imaging of neural activity requires bright fluorescent probes localized in the neuron membrane. Rhodopsins are membrane proteins that can be expressed in neurons and used for triggering, silencing and reporting on neuronal action potentials[1,2]. The prototypical fluorescent reporter is Archaerhodopsin-3 (Arch3), an archaeal rhodopsin from *Halorubrum sodomense*[3–5]. However, the fluorescence of Arch3 is extremely dim; its quantum yield (FQY) of ca. $1.1 \cdot 10^{-4}$ is almost four orders of magnitude lower than that of the green fluorescent protein (GFP)[6–11]. Furthermore, the fluorescence does not come from the dark-adapted state but rather from a photochemically produced photocycle intermediate that cannot deliver a prompt emission signal[11]. These facts not only make single neuron studies impossible with common microscopy techniques, but impair the investigation of neural activity at the population level as wide-field imaging techniques require improved spatial and temporal resolution[12]. In order to achieve better reporters, Arch3 has been engineered via directed evolutionary approaches and random mutagenesis, ultimately discovering variants

such as the Archers[13], the QuasArs[12], the Archons[14], Arch5 and Arch7[13]. Recently, Hegemann and coworkers have investigated new Archon1 variants to elucidate the mechanism of fluorescence voltage sensitivity[15]. These variants feature a higher $10^{-3}$–$10^{-2}$ FQYs enabling, among other applications, the imaging of neuronal activity in living mammals and invertebrates[16–18]. It has also been suggested that such enhanced fluorescence originates, in contrast to Arch3, from a one-photon electronic excitation and, therefore, must come from the protein dark-adapted state[12,14,15]. Here we report on a (mechanistic) theory that explains how one-photon FQY is enhanced in Arch3 variants. We demonstrate that, to be predictive, such theory requires the mapping of an isomerization path producing a twisted intramolecular diradical intermediate (TIDIR) located in proximity of a conical intersection (CoIn). The energy difference between TIDIR and the emissive planar fluorescent state (FS), or $\Delta E_{TIDIR-FS}$, controls the magnitude and positions of the $S_1$ photoisomerization barrier ($E^f_{S_1}$) that, in turn, determines the FQY.

[1]Dipartimento di Biotecnologie, Chimica e Farmacia, Università di Siena, via A. Moro 2, I-53100 Siena, Italy. [2]University of Durham, Department of Chemistry, South Road, Durham DH1 3LE, United Kingdom. [3]Department of Chemistry, Bowling Green State University, Bowling Green, OH 43403, USA. [4]Institut de Chimie Radicalaire (UMR-7273), Aix-Marseille Université, CNRS, 13397 Marseille, Cedex 20, France. [5]Department of Chemistry, Kyungpook National University, Daegu 702-701, South Korea. [6]University of Strasbourg Institute for Advanced Studies, 5, alleé duGeń eŕ al Rouvillois, F-67083 Strasbourg, France. [7]Present address: Centre for Computational Chemistry, School of Chemistry, University of Bristol, Bristol, BS8 1TS, United Kingdom. [8]Present address: Dipartimento di Chimica e Chimica Industriale, Università di Pisa, Via Giuseppe Moruzzi, 13, I-56124 Pisa, Italy. ✉e-mail: olivucci@unisi.it

Based on the above result, we show that the increased FQY displayed by certain Arch3 variants is determined by inhibiting an electrostatic barrier suppression mechanism operating in the naturally occurring progenitor.

## Results and discussion

### Arch photoisomerization mechanism and $S_1$ PES topography

The function of rhodopsins is triggered by the photoisomerization of their retinal chromophores (Fig. 1A) that also operate as fluorophores[19]. Consistently with previous studies on fluorescent proteins[20], we hypothesize that the competition between isomerization and light emission determines the FQY value (see Supplementary Methods 3). This can be understood by examining a schematic representation of the chromophore excited state isomerization path (Fig. 1B) leading to a CoIn between the excited ($S_1$) and the ground ($S_0$) states where excited state decay occurs. From the scheme it is evident that the isomerization barrier ($E^f_{S_1}$) located along the potential energy surface of the spectroscopically allowed $S_1$ state is a critical quantity. The higher $E^f_{S_1}$ is, the slower the isomerization is. Consequently, paths with high $E^f_{S_1}$ will feature an approximately planar fluorescent state FS with long $S_1$ lifetimes and, thus, high FQY. Below, we demonstrate a proportionality between *computed* $E^f_{S_1}$ and *observed* FQY by constructing multi-configurational quantum-chemical based models (MCQC) of the dark-adapted state of a set of Arch3 variants called the Arch-set (Supplementary Methods 1–3 and Supplementary Figs. 1–3). We will also show that the proportionality between *computed* $E^f_{S_1}$ and *observed* FQY remains valid when $E^f_{S_1}$ is replaced by the isomerization energy $\Delta E_{TIDIR-FS}$. Our MCQC models employ the well-established CASSCF[21] zeroth-order wavefunction defined by the selection of a

(12,12) active space including all the π-electrons and orbitals of the retinal chromophore. Although the trends in spectral properties discussed throughout the text are well reproduced at this level, we discuss the results obtained after multi-state (XMS-CASPT2)[22,23] energy and geometrical corrections to the CASSCF geometries.

Before focusing on the relationship between $E^f_{S_1}$ and FQY, it is useful to describe the changes of the all-*trans* retinal chromophore of Archaearhodopsins[24] along the isomerization path computed using the constructed Arch3 model. It is established that at least two geometrical coordinates are implicated in rhodopsin $S_1$ isomerization coordinates (Fig. 1C)[25–27]. The first describes the chromophore initial relaxation from the Franck-Condon (FC) point and corresponds to a bond length alternation (BLA) stretch. The second is the twisting (α) of the reactive *trans* $C_{13} = C_{14}$ bond ultimately leading to the 13-*cis* configuration. As shown in Fig. 2A and consistently with the scheme of Fig. 1B, such relaxation leads to the potentially emissive FS intermediate. This process is followed by progression along a flat potential energy region characterized by a monotonic decrease in α and connecting FS to a TIDIR intermediate located close to a sloped[28] CoIn. As it will be explained below, this previously unreported intermediate differs, in terms of electronic structure and topography, from the locally excited (LE) intermediate identified in a ring-locked derivative of bovine rhodopsin by Laricheva et al.[29].

### Electronic character of the $S_1$ PES

In Fig. 2B we show that such a progression is replicated by the MCQC model of the top fluorescent variant Arch7. However, the comparison between the Arch3 and Arch7 $S_1$ energy profiles shows a significantly increased $E^f_{S_1}$ in the latter. Such an increase is seen in both the zeroth-order wavefunction calculation and in the more quantitative profiles obtained after applying geometrical and multi-state second-order perturbative correction (Supplementary Methods 4 and 5, Supplementary Figs. 4–7). Due to the high computational cost of QM/MM analytical Hessians, the TSs discussed throughout the text are approximated by the energy maxima along the relaxed scan connecting FS and TIDIR. These TSs must be considered approximate as it has not been possible to carry out a geometry optimization starting from a computed Hessian matrix as well as to compute a Hessian matrix at the end of the TS search. Furthermore, despite the increased FQY of the investigated Arch variants, recent measurements of the excited state lifetime (ESL) of some of the variants were found to be in the time range of picoseconds[30,31]. For this reason, we don't account in our calculation for $T_1/S_1$ intersystem crossing (ISC) as a viable competitive process to $S_1$ emission also considering that $T_1$ is a π-π* state with orbitals parallel (non-orthogonal) to those characterizing the $S_1$ state. Therefore, the singlet to triplet transition would be "forbidden" by the El-Sayed rule.

The progression along α is documented in Fig. 2C, D where we report the values of the relevant geometrical parameters of FS and TIDIR as well as the charge residing on the Schiff base moiety $C_{14}$-$C_{15}$-N. In fact, the geometrical changes are accompanied by variations of the chromophore electronic character. These can be conveniently followed by computing the fraction of positive charge and free valence (NUE)[32] residing on $C_{14}$-$C_{15}$-N (Supplementary Methods 6 and Supplementary Figs. 4, 7, 9 and 10). Following a recent report on the variants of DRONPA2[20], a soluble GFP-like protein, we interpret the results in Fig. 2A, B in terms of mixing of two diabatic states describing the covalent ($1A_g$) and charge-transfer ($1B_u$) characters of polyenes[33]. The computed $C_{14}$-$C_{15}$-N charge and NUE progression point to an $S_1$ state initially dominated by the $1B_u$ charge transfer character as already reported for other rhodopsins. However, such character decreases when relaxing to an FS that features a larger $1A_g$ weight, then constantly increasing all along the isomerization path. In summary, the electronic character evolves from mixed charge-transfer/covalent $1A_g/1B_u$ characters at FC and FS, to a substantially pure $1A_g$ character at

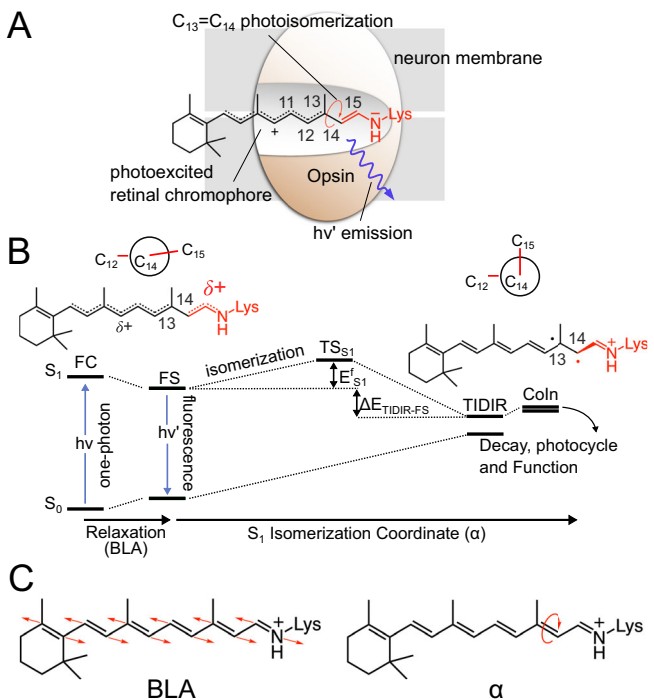

**Fig. 1 | Photoisomerization mechanism of Archaerhodopsins. A** Lewis formula representing the initial $S_1$ chromophore structure. **B** Representation of the chromophore isomerization path. FS corresponds to the fluorescent state. TIDIR represents the photoisomerization channel located near CoIn. FS and TIDIR are represented by Lewis formulas displaying distinct degrees of double bond twisting and charge transfer. **C** Main components of the reaction coordinate. BLA is numerically defined as the difference between the average single-bond length minus the average double-bond length along the C5 to N conjugated chain (for convenience, below we consider the BLA of the framed moiety exclusively). α is defined by the dihedral angle C12-C13-C14-C15.

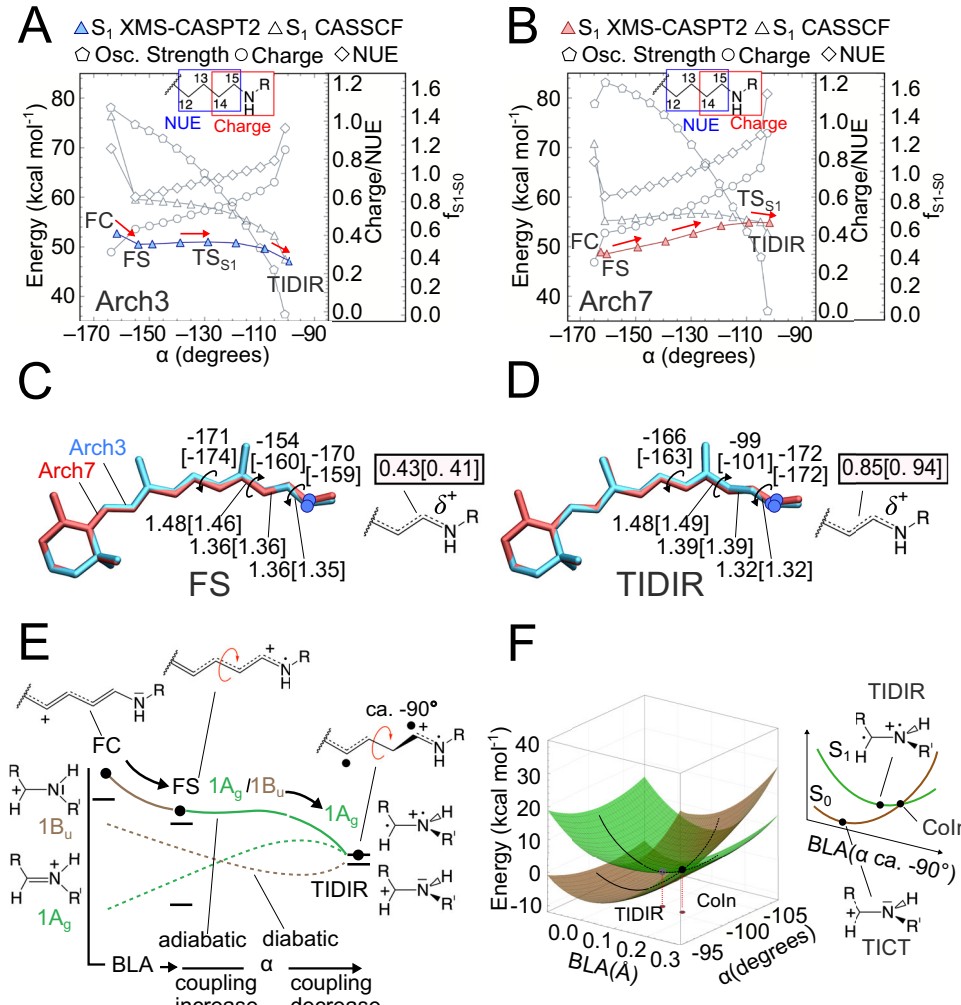

**Fig. 2 | Comparison between computed Arch3 and Arch7 $S_1$ isomerization paths. A** Variations in potential energy (relative to the $S_0$ equilibrium structure), charge distribution, free valence and oscillator strength along the Arch3 path. The energy profile in color is given after MS correction. Red arrows indicate progression along the isomerization coordinate. **B** Same data for Arch7. **C** Main geometrical chromophore parameters for the FS fluorescent intermediates of Arch3 and Arch7 (values in square brackets). The $S_1$ positive charge fraction on the C-C-N moiety are also given. **D** Same data for the photoisomerization channel TIDIR. **E** Schematic "decomposition" of the Arch3 adiabatic energy profile of panel **B** in terms of diabatic states associated to Lewis formulas of the $CH_2 = NH_2(+)$ minimal model. The progression of the electronic and geometrical structure of the relevant moiety of the full chromophore is given at the top. **F** Schematic representation of the CoIn region of Arch3 including the twisted diradical TIDIR along the relevant components of the reaction coordinate. The same coordinate also spans the CoIn branching plane. The potential energy scale is relative to the CoIn.

TIDIR that features an α value of ca. −90°, two spin-paired, but non-interacting, radical centers located on two orthogonal π-systems and a positive charge fully confined on $C_{14}$-$C_{15}$-N. This process is accompanied by a change in the $S_1$-$S_0$ oscillator strength ($f_{S1-SO}$) along α, from values typical of allowed electronic transitions to a forbidden transition at TIDIR.

The evolution going from a mixed $1A_g/1B_u$ character at FS to a $1A_g$ covalent/diradical character at TIDIR, provides information on the origin of the $E^f_{S1}$ barrier. As illustrated in Fig. 2E the electronic coupling between the $1A_g$ and $1B_u$ diabatic states (represented, for simplicity, by the Lewis formula of a methylimine cation model) would initially increase due to the decrease in their energy gap. However, when α approaches orthogonality, the coupling decreases rapidly and becomes negligible at TIDIR. The negligible coupling at TIDIR is related to the vicinity of a CoIn (see Fig. 2F) where the $1A_g/1B_u$ coupling is zero. A CoIn deformation along the negative BLA direction lifts the degeneracy and intercepts TIDIR (see Supplementary Methods 7 and Supplementary Figs. 11 and 12). Notice that the same deformation along the $S_0$ potential energy surface achieves the

twisted intramolecular charge transfer TICT structure[34] corresponding to the transition state driving the chromophore thermal isomerization also documented for bovine rhodopsin[33]. In conclusion, we associate the variation in the magnitude and position of $E^f_{S1}$ (i.e., of $TS_{S1}$ in Fig. 1B) with the variation along α of the $1A_g$ and $1B_u$ diabatic energies and their $1A_g/1B_u$ coupling.

## $S_1$ isomerization barrier determines an increased FQY in the Arch variants

We now show that the isomerization mechanism above provides the basis for understanding the FQY variations along the Arch-set. To do so, we first demonstrate the existence of a correlation between computed $E^f_{S1}$ and observed FQY values and then rationalize it based on FS and TIDIR charge distributions (or $1A_g$ and $1B_u$ character) calculated by constructing the MCQM models of all variants and using them to map the progression along α. The model accuracy is documented in Fig. 3A where we compare, after multi-state perturbative correction, the computed and observed trends of absorption ($\Delta E^a_{S1-SO}$, top) and emission ($\Delta E^f_{S1-SO}$, bottom) vertical excitation

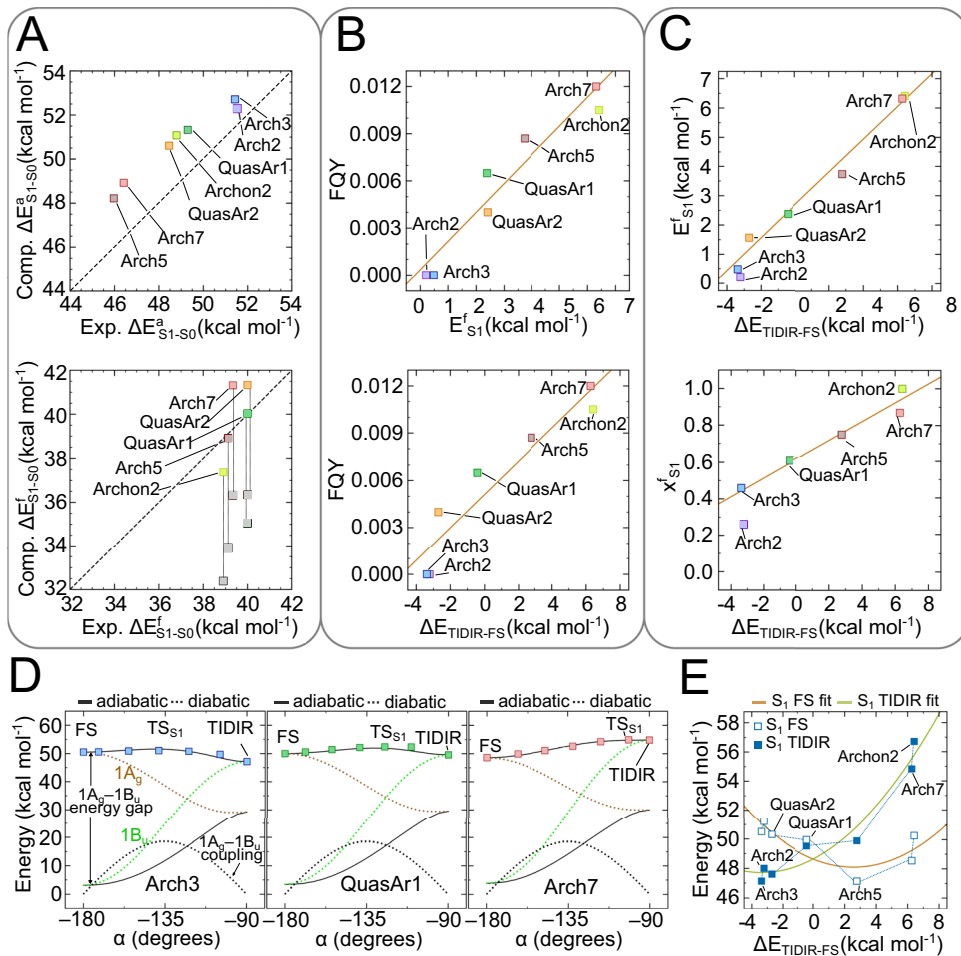

**Fig. 3 | Relationship between computed $S_1$ isomerization properties and observed FQY along the Arch-set. A** Comparison between computed and observed excitation energies associated with light absorption at the ground state equilibrium structure (top) and light emission from the fluorescent state (bottom). Vertical excitation energy associated to emission features a correction of 4.5 kcal mol$^{-1}$ to account for kinetic energy derived from ref. [24]. Gray and colored squares represent the emissions before and after correction, respectively. **B** Relationship between $S_1$ isomerization barrier and FQY (top) and between $S_1$ isomerization energy and FQY (bottom). Linear fits are given as orange lines. **C** Relationship between $S_1$ isomerization barrier and isomerization energy (top) and between $S_1$ isomerization transition state position and isomerization energy for three Arch3 variants. Linear fits are given as orange lines. **D** One-dimensional model for the relationship between barrier and isomerization energy. **E** Non-linear relationship between the FS and TIDIR $S_1$ excitation energies with respect to $S_0$ FC and the isomerization energy. Source data are provided in the Source Data file (d).

energies. The analysis of the results demonstrates that, in all variants, FS and TIDIR feature the same distinct charge distributions (Supplementary Figs. 4 and 7).

The bottom panel of Fig. 3B shows that computed $E^f_{S1}$ and observed FQY variations are directly proportional. If one assumes that the Hammond-Leffler postulate[35,36] is valid for an $S_1$ isomerization, $E^f_{S1}$ and $\Delta E_{TIDIR-FS}$ (i.e., the reaction endothermicity or, simply, isomerization energy in Fig. 1B) must also correlate. This leads to the conjecture that $\Delta E_{TIDIR-FS}$ is proportional to FQY. The bottom panel of Fig. 3B demonstrates that such proportionality exists. In other words, a progressive stabilization of the "reactant" FS and/or destabilization of the "product" TIDIR, must lead to higher FQYs. The relationship between the computed $E^f_{S1}$ and $\Delta E_{TIDIR-FS}$ for the entire Arch-set can be modeled by using a basic two-state one-mode Hamiltonian that fits the computed $S_1$ and $S_0$ energy profiles along mode α in terms of $1A_g$ and $1B_u$ diabatic energies and a harmonic $1A_g/1B_u$ coupling function (Supplementary Methods 8 and Supplementary Fig. 12). The maximum along the resulting $S_1$ energy profile represents $TS_{S1}$ and, as demonstrated in Fig. 3D for Arch3, QuasAr1 and Arch7, replicates the computed MCQC $S_1$ energy profiles. As shown in Fig. 3C the model Hamiltonian supports the existence of a proportionality between $E^f_{S1}$ (top panel) and the $TS_{S1}$ position $X^f_{S1}$ (bottom panel) and $\Delta E_{TIDIR-FS}$. We stress that this

mechanistic model assumes no "a priori" relationship between diabatics energy difference and diabatic coupling, but is a simple valence-bond type description of our adiabatic $S_1$ PES, assuming two pure resonance formulas ($1A_g$ and $1B_u$), whose weights are associated to the documented variation in positive charge distribution along the reaction path.

The $\Delta E_{TIDIR-FS}$ increases along the Arch-set may originate from either FS stabilization or a TIDIR destabilization effects (or from both effects). In Fig. 3E we show that the first effect (i.e., substantially the 0-0 excitation energy) shows a decrease until Arch5 that has the lowest excitation energy and then increase up to Archon2. In contrast, the TIDIR destabilization with respect to the same reference (i.e., the destabilization of the photoisomerization channel) increases almost monotonically indicating a higher sensitivity of FQY to mutations changing $\Delta E_{TIDIR-FS}$ rather than the difference in energy between FC and FS. The described $\Delta E_{TIDIR-FS}$ changes are confirmed by probing the light-induced dynamics of the Arch-set. To do so we employed the variants MCQC models to propagate a quantum-classical trajectory released from the FC point with zero initial nuclear velocities (Supplementary Methods 9 and Supplementary Figs. 14 and 15). As discussed in the literature, these trajectories mimic the evolution of the center of the $S_1$ population and are useful to detect barrierless or

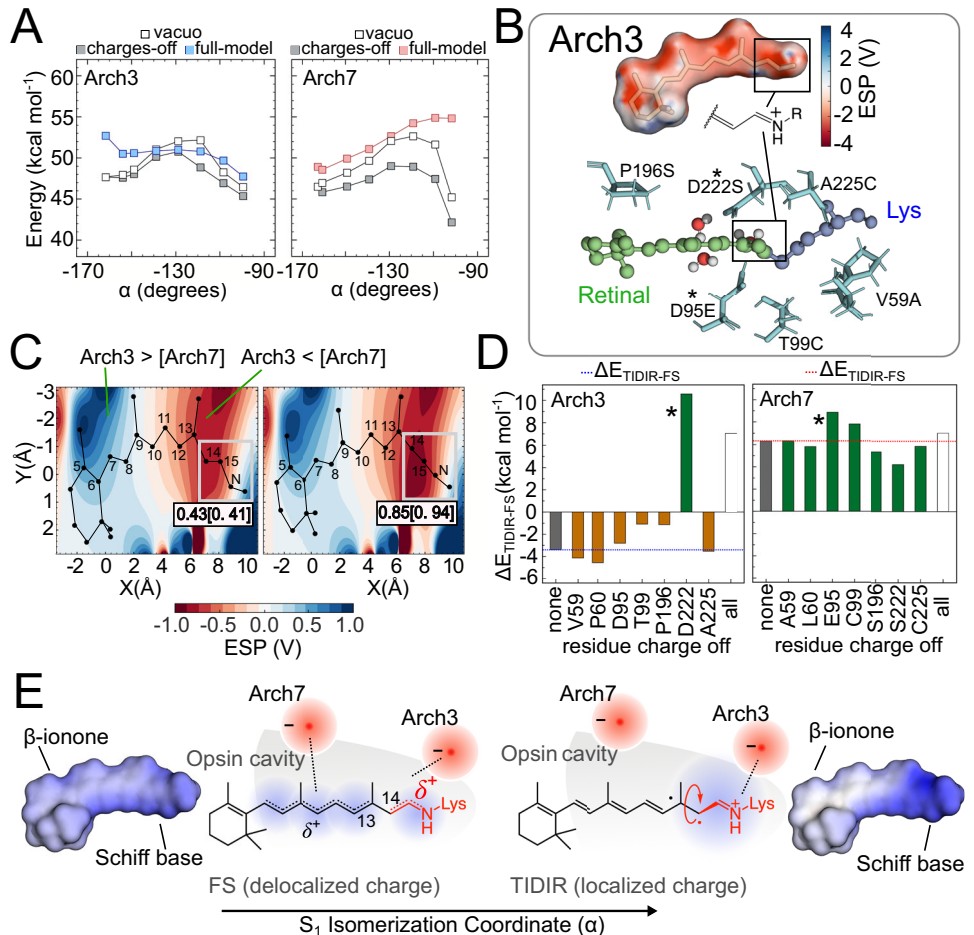

**Fig. 4 | Origin of the $S_1$ isomerization barrier in Arch3 and Arch7. A** Effect of the opsin charges on the isomerization energy profile for Arch3 (left) and Arch 7 (right). **B** Top. Arch3 opsin electrostatic potential projected on the chromophore solvent accessible surface. Bottom. The cavity amino acids distinguishing Arch3 from Arch7. The * symbol indicates the Arch3 (D222) and Arch7 (E95) chromophore counterions. **C** Two-dimensional plots showing the Arch3-Arch7 electrostatic potential difference at FS (left) and TIDIR (right). The total charge of the framed Schiff-base moiety is also given. **D** Variation in the Arch3 (left) and Arch7 (right) isomerization energy value after zeroing the residue charge of the cavity residues in panel **B** bottom. Reference values are shown as gray bars. Negative and positive isomerization energies are shown in orange and green, respectively. Finally, the effect of zeroing the charges of all residues simultaneously is also given as an open bar. **E** Point charge model for the delocalization-confinement mechanism. The Arch3-Arch7 ESP difference of **C** can be modeled in terms of a re-locating effective negative charge from the Schiff base region to a cavity region displaced towards the β-ionone ring of the chromophore. The increase in distance between the effective charge and the confined positive charge of TIDIR explains its electrostatic destabilization in Arch7 with respect to Arch3. The computed chromophore electrostatic potential projected on the chromophore solvent accessible surface is given at the left and right of the figure to support the schematic representation of a delocalized and confined charge given at the center. Source data are provided in the Source Data file (a, c, d).

nearly barrierless isomerization paths and validate the accuracy of computed torsional scans[27,37].

Consistently with the trend in $E^f_{S1}$ values, the Arch3 trajectory reaches the CoIn region and decays to $S_0$ on a sub-500 fs timescale via a $C_{13} = C_{14}$ isomerization pointing to species that do not display significant fluorescence. In contrast, the QuasAr1 and Arch7 trajectories representing the remaining variants, orbit in the FS region for the entire 450 fs simulation time.

**Molecular determinants of the isomerization barrier**

Above we have demonstrated that our MCQM models produce a $\Delta E_{TIDIR-FS}$ trend proportional to the observed FQYs for the entire Arch-set. We now focus on the limiting cases of Arch3 and Arch7 to show that the proportionality between $\Delta E_{TIDIR-FS}$ and FQY is linked to the variations in the protein (opsin) sequence. This is possible because, as seen in Fig. 2C, D, the FS and TIDIR retinal chromophores have distinct charge distributions and, therefore, $\Delta E_{TIDIR-FS}$ must be sensitive to the protein electrostatics. In other words, an opsin electrostatic potential ($ESP_{opsin}$) stabilizing FS or/and destabilizing TIDIR would produce a

larger $\Delta E_{TIDIR-FS}$ value and, in turn, enhance FQY. Such effect has been investigated by recomputing the $S_1$ energy profiles along α after setting to zero the opsin charges of the MCQC models (Supplementary Methods 10 and Supplementary Figs. 15–17). It is apparent from inspection of the energy profiles in Fig. 4A, B that, in the absence of the protein electrostatics, both Arch3 and Arch7 display a sizable energy barrier along α. However, the left panel of Fig. 4A demonstrates (compare the full model with the charges-off energy profiles) that, in Arch3, the opsin charges stabilize TIDIR with respect to FS yielding a low $\Delta E_{TIDIR-FS}$ value and, consequently, a negligible $E^f_{S1}$. In contrast, the right panel of Fig. 4A shows that, in Arch7, the opsin charges have the opposite effect. We can conclude that in the absence of the protein electrostatics, the geometrical deformation imposed by the opsin cavity on the chromophore backbone leads to a sizable $E^f_{S1}$ value. Notice that the geometrical deformation is the result of both cavity steric and electrostatic effects on the chromophore isomerization coordinate and that these effects are common to Arch3 and Arch7, as well as to all members of the Arch-set (Supplementary Methods 10). This has been confirmed by recomputing the same energy profile in

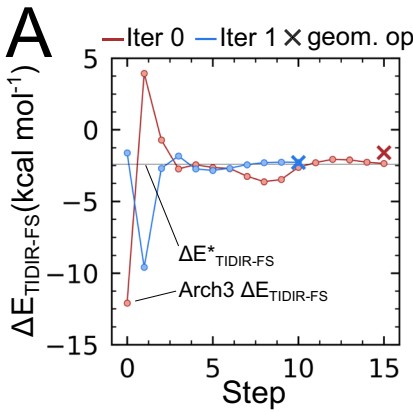

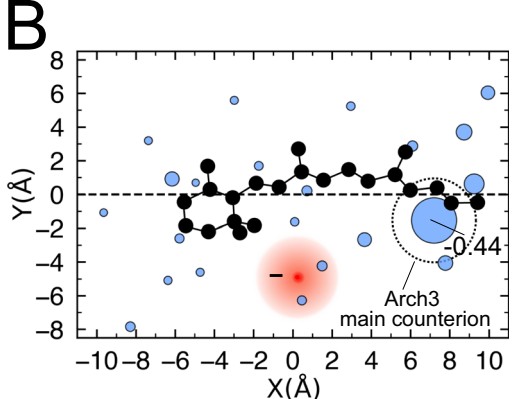

**Fig. 5 | Predicted change of counterion charge distribution of Arch3 QM/MM model to yield the Arch7 $\Delta E_{\text{TIDIR-FS}}$ value. A** $\Delta E_{\text{TIDIR-FS}}$ (SA2-CASSCF/AMBER level) change along the optimization steps leading from the Arch3 to the Arch7 value. The optimization is driven by the target $\Delta E_{\text{TIDIR-FS}}$ values ($\Delta E^*_{\text{TIDIR-FS}}$, indicated by a thin horizontal line) and corresponds to the minimization of the square of the scalar function $\Delta E_{\text{TIDIR-FS}}(q) - \Delta E^*_{\text{TIDIR-FS}}$ (i.e., $\Delta\Delta E_{\text{TIDIR-FS}}(q)$) as a function of the cavity residue charge vector q. Each iteration of the algorithm (two full iterations are reported in the panel) is divided in two parts; (i) first q is optimized at fixed FS and TIDIR geometries via conjugated-gradient optimization such that $\Delta E_{\text{TIDIR-FS}}(q) = \Delta E^*_{\text{TIDIR-FS}}$ and then (ii) the FS and TIDIR QM/MM model geometries are relaxed, and

$\Delta E_{\text{TIDIR-FS}}$ is recomputed. If the difference between the $\Delta E_{\text{TIDIR-FS}}$ calculated after part ii and $\Delta E^*_{\text{TIDIR-FS}}$ is above a selected threshold, i-ii are repeated. The circles indicate the steps of part i, and the crosses indicate the results of geometrical relaxation of part ii. **B** Final charge distribution obtained after convergence of the optimization above. Two-dimensional representation of the fraction of negative charges residing in the cavity residues is proportional to the radius of the blue circles (the main counterion D222 final charge is indicated). The barycenter of the corresponding negative charge is shown as a red circle. The original localized Arch3 negative charge is indicated by the large open circle centered at the original counterion position Source data are provided in the Source Data file (a, b).

vacuo (i.e., in absence of the VdW interaction between protein cavity and chromophore) that still show a barrier in all cases.

The described electrostatic stabilization of TIDIR relative to FS in Arch3, and the consequent disappearance of the barrier, has been rationalized by mapping the ESP$_{\text{opsin}}$ on the chromophore solvent accessible surface (Fig. 4B, top). The map shows a prevalent negative ESP$_{\text{opsin}}$ value in the area surrounding the Schiff base moiety whose effect along the isomerization coordinate is described by plotting the difference ($\Delta$ESP$_{\text{opsin}}$) between the Arch3 and Arch7 electrostatics. In fact, the $\Delta$ESP$_{\text{opsin}}$ mapped along a cross-section roughly parallel to the FS (Fig. 4C, left) and TIDIR (Fig. 4C, right) chromophore backbones, demonstrates that the opsin of Arch3 preferentially stabilizes the positive charge fully confined on the Schiff base moiety of TIDIR while the opsin of Arch7 better stabilizes the delocalized charge spread towards the β-ionone of FS. These effects can be also discussed using the model Hamiltonian presented above where the 1B$_\text{u}$ diabatic dominates the FS while the 1A$_\text{g}$ diabatic dominates the TIDIR. Since a Schiff base confined charge is a characteristic of 1A$_\text{g}$, the Arch3 electrostatics must stabilize the covalent/diradical 1A$_\text{g}$ character at TIDIR while the Arch7 electrostatics destabilizes it consistently with the diabatic model of Fig. 3D. It is possible to demonstrate that this effect remains valid when one uses a more realistic two-state two-mode Hamiltonian model generating two-dimensional adiabatic energy surfaces function along both BLA and α (Supplementary Methods 8).

## Role of the main counterion in the barrier generation or suppression

At this point we propose an atomistic mechanism for the Arch3 barrier suppression. The differences in the ESP$_{\text{opsin}}$ of Arch3 and Arch7 is a product of the protein sequence variation and, more specifically, from the seven residue replacements displayed at the bottom of Fig. 4B. In our MCQC models, the ESP$_{\text{opsin}}$ is produced by the point charges centered on the atoms of each protein residue (Supplementary Methods 2). Therefore, the $\Delta$ESP$_{\text{opsin}}$ of Fig. 4C must reflect the difference in point charges before and after the residue replacements. The effect of such difference has been investigated by recomputing $\Delta E_{\text{TIDIR-FS}}$ after setting the point charges of residues 59, 60, 95, 99, 196, 222, 225 as a group or individually. The results in Fig. 4D show that when setting to zero the charges of all residues, the $\Delta E_{\text{TIDIR-FS}}$ of Arch3

and Arch7 become similar. More specifically and consistently with the results in Fig. 4A, B, the Arch3 point charges must cause a ca. 9 kcal mol$^{-1}$ decrease in $\Delta E_{\text{TIDIR-FS}}$ while the corresponding Arch7 residues do not seem to affect the original ca. 6 kcal mol$^{-1}$ $\Delta E_{\text{TIDIR-FS}}$ value. From the analysis of the individual residues, it is apparent that the counterion at position 222 dominates the Arch3 electrostatics. Indeed, during the Arch3 to Arch7 transition the counterion is moved from position 222 to position 95 while the position 222 is taken by an uncharged cysteine. In conclusion, the counterion relocation appears to be the main mechanism for the $\Delta E_{\text{TIDIR-FS}}$ modulation. This is not a general mechanism. In fact, other members of the Arch-set all have the counterion in position 222 (Supplementary Methods 1) but display a regularly increasing $\Delta E_{\text{TIDIR-FS}}$ value. In these cases, the effect of the residue replacements must cause a ESP$_{\text{opsin}}$ change that partially screen the counterion effect found in Arch3, which may be interpreted in terms of a "virtual relocation" of the counterion as explained below.

We propose that this relocation and "diffusion" of the negatively charged counterion (a virtual counterion), coupled with a delocalization-then-confinement mechanism of the positive charge of the chromophore, explains the regular change in isomerization barriers. In brighter Arch-3 variants (Arch-5 and Arch-7) the virtual counterion is increasingly distant and more diffuse from the Schiff base moiety. In these variants, at α = 0° the chromophore positive charge is delocalized, and its centroid is close to the counterion, leading to a stabilization of FS. As soon as α progresses and approaches a 90° twist, the confinement of the chromophore charge on the Schiff base moiety gradually increases, causing its centroid to drift away from the virtual counterion, inevitably determining a de-stabilization of TIDIR.

To support this conclusion, we developed a basic model that allows to compute (i.e., optimize) the protein charge distribution inducing a specific $\Delta E_{\text{TIDIR-FS}}$ value. This is done by allowing the relocation and fragmentation (i.e., diffusion) of the negative charge of the main counterion to other cavity residue positions, to mimic what observed in Fig. 4E. This can be achieved by defining a scalar function of the cavity residue charge vector (q) which returns $\Delta E_{\text{TIDIR-FS}}$. Such $\Delta E_{\text{TIDIR-FS}}(q)$ function is differentiated numerically to study how $\Delta E_{\text{TIDIR-FS}}$ responds to q. As detailed in Supplementary Methods 11, the problem of determining q can be formulated as a constrained optimization. In Fig. 5, we show how the optimization modifies the charge

distribution of Arch3 to reproduce the $\Delta E_{TIDIR-FS}$ value computed for Arch7 at the zeroth-order level. The resulting q shows that, for Arch3 to reproduce Arch7 excited state properties, a relocation of ca. 50% of D222 negative charge to other cavity residues is necessary, thus supporting the conclusion that a relocation and diffusion of the counterion is indeed a determinant of the TIDIR destabilization in the Arch-set.

## Conclusions

The engineering of efficient, rhodopsin-based action-potential reporters (also called Genetically Encodable Voltage Indicators or GEVIs in the literature) is a challenge currently addressed by experimentally investigating the fluorescence and voltage sensitivity mechanisms[15] and by tuning the protein fluorescence via several rounds of random mutagenesis and/or directed evolution. Above we have investigated the increase in fluorescence intensity in a set of Arch3 variants using MCQC models with second order perturbative corrections. The key results are: (i) the FQY is primarily determined by the competition between $S_1$ emission and isomerization extending the mechanism reported for DRONPA2[20] to retinal proteins, (ii) the isomerization rate is governed by the stability of the rather exotic TIDIR intermediate with respect to the fluorescent state, (iii) the TIDIR stability is modulated by the electrostatics of the protein that increasingly offset the $S_1$ isomerization energy barrier when going from Arch7 to Arch3.

The atomistic mechanism for the TIDIR stabilization appears to be a consequence of the cavity electrostatics on the distinct charge distributions of the FS and TIDIR chromophores. As schematically (top schemes) and computationally (bottom plots) displayed in Fig. 4E, the chromophore positive charge is largely delocalized in the FS but confined on the $C_{14}$-$C_{15}$-N fragment in the TIDIR. In this situation the described change in electrostatics from Arch3 to Arch7 leads to a reduced stabilization of the Schiff base confined with respect to the delocalized charge. In simple terms, the effect of the amino acid (and atomic point charges) replacement resulting in the Arch3 to Arch7 progressive TIDIR destabilizing along the Arch-set, can be interpreted as the gradual relocation of a negative charge from the Schiff base region to a region closer to the β-ionone ring.

The presented mechanism has both methodological and biological implications. The first is related to the fact that TIDIR and FS, being energy minima on the $S_1$ potential energy surface, are computationally fast to locate. It is thus possible to envision the development of a tool for the in silico selection of highly fluorescent Arch3 variants. Such tool would be based on the automated[38] construction of MCQC models and the calculation of the corresponding $\Delta E_{TIDIR-FS}$ value to be maximized via in silico mutational experiments. The biological implication is instead related to the hypothesis that the changes in sequence, and thus electrostatics, leading to a negligible Arch3 fluorescence could have occurred also in nature through a natural selection process aimed at increasing the protein photoisomerization rate and improve the protein function. We cannot exclude that the same mechanism plays a role in other microbial rhodopsin evolution. On the other hand, an inverse natural selection process leading to the suppression of the photoisomerization, and maximization of the fluorescence output may have generated the recently discovered Neorhodopsin from *Rhizoclosmatium globosum*[39] that displays an intense fluorescence with a ca. 0.2 FQY and therefore much closer to that of DRONPA2 and other optimized green fluorescent protein variants.

## Methods

The three-dimensional structures of the Arch-set proteins were obtained either from the corresponding crystallographic structures deposited in the Protein Data Bank (Arch2 and Arch with PDB ID:3WQJ and 6GUX, respectively) or via the comparative model (the Arch3 variants) approach described in Supplementary Methods 1. The corresponding QM/MM models were constructed from the obtained three-dimensional structures according to the *a*-ARM based protocol[38] discussed in Supplementary Methods 2 with the QM and MM subsystems interacting via an electrostatic embedding scheme. The retinal chromophore atoms, the chromophore-bound lysine side chain atoms starting from the N-terminal to the Cε and the hydrogen link atom (HLA) are included in the QM subsystem and treated at the multiconfigurational (CASSCF) level, while the rest of the atoms of the model are treated at the MM level using the AMBER94 force field[40].

The ground state optimized geometries are obtained via geometry optimization at the single-state CASSCF(12,12)/6-31G*/AMBER level of theory. Further refinement was performed introducing a perturbative geometrical correction to produce XMS-CASPT2/SA3-CASSCF(12,12)/ANO-L-vDZP/AMBER geometries (Supplementary Methods 4 and 5). Both the CASSCF and XMS-CASPT2 models were then used to compute reaction paths along the isomerization coordinate via a relaxed scan taking, as the reacting coordinate, the twisting of the reactive double bond. The CASSCF geometries were also employed to probe the excited state dynamics of three models via non adiabatic deterministic surface-hop trajectories propagated from the Franck-Condon (FC) point on $S_1$ with zero initial velocities (FC trajectories) at the SA2-CASSCF(12,12)/6-31G*/AMBER level using the Tully algorithm[41]. All the QM/MM calculations were performed using the [Open]Molcas v19.11/TINKER and MOLCAS v8.1/TINKER packages[42–44].

## Data availability

Cartesian coordinates of the FC, FS, CoIn and TIDIR geometries of the QM/MM models calculated at the SA2-CASSCF(12,12)/6-31 G*/AMBER level of theory for the models of the Arch set are provided as Supplementary Data 1.

Cartesian coordinates of the FC, FS and TIDIR geometries of the QM/MM models calculated at the XMS-CASPT2/SA3-CASSCF(12,12)/ANO-L-vDZP/AMBER level of theory for the models of the Arch set are provided as Supplementary Data 2. Source data is provided with this paper. Source data are provided with this paper.

## Code availability

All the calculations discussed in this work are based on the MOLCAS/TINKER interface.

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

## Acknowledgements

The research has been partially supported by the following grants: NSF CHE-CLP-1710191, NIH GM126627-01, USIAS 2015, the Ohio Supercomputer Center, the MIUR (Ministero dell'Istruzione, dell'Università e della Ricerca) for a "Dipartimento di Eccellenza 2018-2022" and the Fondazione Banca d'Italia to M.O. The MIUR is also acknowledged for a Rita Levi Montalcini grant to D.P. N.F. and M.H.-R. acknowledge the financial support by Agence Nationale de la Recherche (project ULTRA-rchea, grant ANR-21-CE11-0029-03). L.B. and M.O. acknowledge partial support from European Union, Next Generation EU, MIUR Italia Domani Progetto mRNA Spoke 6 del "National Center for Gene Therapy and Drugs based on RNA Technology". CUP di progetto B63C22000610006. The authors are grateful to Xuchun Yang and María del Carmen Marín for fruitful discussion.

## Author contributions

Conceptualization and Project administration: M.O., Methodology: L.P.G., E.M., L.B., D.P., L.D.V., N.F., M.H.R., M.F., D.K., A.B.G., Investigation and Visualization: L.B., E.M., M.O., D.K., Funding acquisition: M.O., D.P., Supervision: D.P., L.D.V., M.O., Writing and Reviewing: M.O., L.B.

## Competing interests

The authors declare no competing interests.
