## [Peer Review File · Nature Communications]

Reviewers' comments:

Reviewer #1 (Remarks to the Author):

The authors tried to explain the experimentally observed increase in quantum yield on the basis of calculated energy levels. In terms of the quantum chemical approach, they use the MCQC models with second order perturbative corrections. Although they usually emphasize the second order perturbative corrections, the study itself is a typical black-box type poorly reasoned theoretical study. To that extent, the publication of this manuscript appears premature to me.

In general there was not sufficient enthusiasm for the manuscript to be published in Nature Communications. I think that this study will be more suitable for a specialized journal but not for Nature Communications.

Reviewer #2 (Remarks to the Author):

The authors present a computational investigation of fluorescence properties of Rhodopsins. The increase in fluorescence quantum yield is interpreted in terms of the features of the S1 potential energy surface, which are in turn related to the diabatic transition from charge transfer to diradical character when going from the Franck Condon point to the S1/S0 CI. The importance of the electrostatic effects of the amino acids close to the chromophore is clearly pointed out.

The authors should address the following remarks.

1) The authors are urged to give another careful read to the manuscript. In fact, there are several typos; several references and cross-links are missing in the supporting information; in the caption of figure 3, reference is made to a panel H which is not present in the figure, etc..

2) In general, there is no relationship between the diabatic coupling and the energy difference of two diabatic states. Then, the initial increase of the diabatic coupling between the Ag and Bu states when the alpha angles closes (and/or BLA increases)

should not be related to the decrease in the Ag-Bu energy gap. Moreover, it is not clear why the Ag-Bu diabatic coupling should be modelled by a sinusoidal function (section S8), which is actually referred in the main text so as "a harmonic coupling function".

3) According to Fig. 3C, it is postulated that the two diabatic states Ag and Bu crosses along the alpha coordinate going from the FS to the TIDIR point, and the S0 and S1 states exchange their characteristics. The authors should try to verify this assumption. For example, one may rely on the fact that, in this kind of crossing, the line integral of the nonadiabatic coupling vector should be equal to $\pi/2$. Alternatively, some kind of diabatization scheme could be employed.

4) Why the trajectories are started with zero kinetic energy? How it can be excluded that a realistic nonzero value of the starting kinetic energy could wash out the difference in the S1 dynamics of Arch3 and Arch7?

Reviewer #3 (Remarks to the Author):

The work proposed by Barneschi et al. offers (reading the abstract) a highly novel study aiming at rationalizing the fluorescence quantum yield of different Archaeorhodopsins by studying computationally the potential energy characteristics of the electronically bright excited state.

This is indeed a highly relevant topic in the field of optogenetics and, more in general, in the field of theranostics, i.e., the combination of therapeutics and diagnostics.

The computationally applied methods can be considered of high-level, although nowadays almost standard within the community of computational quantum chemists interested in electronically excited states.

The work is well presented, with almost no grammar errors, and offering a systematic rationalization of the findings, thanks to well-designed figures.

Nevertheless, in spite of the abstract promises and potential importance in a multidisciplinary field, the article lacks the required novelty and breakthrough type of result that is expected for a Nature Communications paper. Actually, if the authors continue developing a really new theoretical/computational tool (that was not the case of this manuscript) "for the in silico selection of highly fluorescent Arch3 variants", pointed out as future perspective in this work, then I would

clearly suggest publication in Nat. Comm. Otherwise, at this stage, this work is perfectly suitable for lower impact factor and more technical (computational chemistry or physical chemistry) journals.

More in detail, the attempt to propose a highly novel excited state mechanism ends up being a rather usual (for rhodopsin systems) S1 potential energy surface where (evidently) the presence of more or less stable (in energy) minimum can be related to the fluorescence quantum yield. This is not a surprise: qualitatively, if the energy barrier to overcome such S1 energy barrier is low (within few kcal/mol units) or absent, then no fluorescence and only photoisomerization or internal conversion can be observed; if the energy barrier is high enough, then photoisomerization is hindered and only fluorescence (or phosphorescence) can be expected; if the situation is in between (the barrier is present, but not that high in energy), then one could have in principle both (fluorescence and photoisomerization), finally matching the required optogenetical expectations.

It is especially surprising that some of the authors already published in 2012 about the origin of fluorescence in a rhodopsin model, by using almost the same techniques and level of theory, but did not mention it in the manuscript: Laricheva et al., J. Chem. Theory Comput. 2012, 8, 8, 2559–2563.

Hence, I do not see any big advance in the field. Also, the diradical character of the highly twisted S1 minimum structure raises up the curiosity whether a relatively high spin-orbit coupling value could lead to triplet population, hence making possible phosphorescence, apart from fluorescence. This aspect is not considered, and would actually increase eventually the novelty of the study, at least in the mechanistic point of view.

On the other hand, I do not feel in this case study the necessity to run a single semiclassical trajectory to show that photoisomerization is possible. This could be the case 10 years ago, when CASSCF excited state trajectories were really at state-of-the-art stage, but not today. If a real dynamical study needs to be undertaken, then hundreds of semiclassical trajectories are required to have a statistical average of the predicted quantum yields (if one rules out quantum dynamics, due to the size of the system).

Also, more at a technical level, I did not fully get (also reading the supporting information) how S1 minima and transition states were obtained. I would remind that such structures do require the calculation of the Hessian matrix, to finally check if all frequencies are positive (minimum) or if one single imaginary frequency is present (transition state). If the authors cannot clarify this issue, their structures are “only” approximated stationary points.

Finally, the authors corrected the CASSCF calculations by introducing the missing electronic dynamic correlation at CASPT2 level through the identification of a BLA “correction vector”. Wouldn't that be simpler to run CASPT2/MM optimizations, nowadays accessible at relatively low computational cost

by using e.g. the BAGEL software? In this way, they could also understand the feasibility of their CASSCF/MM approach, especially considering that few kcal/mol could completely change the picture in this case.

All in all, I feel that this is a nice work, but too preliminary to be published. If the mentioned technical comments can be addressed, I recommend publication in a more specialized journal.

Minor points:

-In the introduction, “four order of magnitude” should be modified as “four orders of magnitude”.

-In Figure 1, the retinal chromophore is covalently linked to ϵ in panels A and B, but then it is linked to R in panel C. This is misleading. The specific protein residue should be instead considered.

-In Figure 2, the legend of panels A and B is insufficient to have an easy readability. Why are there two energy curves? Are they referred to S1 and S2?

Point-by-point reply to the reviewers and list of changes.

Reviewer comments in black, author response in blue and changes in red.

Reviewer #1 (Remarks to the Author):

Main points raised by the reviewer:

"... Although they usually emphasize the second order perturbative corrections, the study itself is a typical black-box type poorly reasoned theoretical study. To that extent, the publication of this manuscript appears premature to me..."

We respectfully, but firmly, disagree with the reviewer. The CASSCF method is not a "black-box" method. On the contrary, the CASSCF method is taken as a textbook example of a "non-black-box" method when compared to, for instance, TD-DFT calculations. The key here is the selection of the active space that is performed by the user based on chemical reasoning (the selection of the state averaging level and level-shift are other decision that the user must take) that, ultimately, defines the CSF-based electronic wavefunction used in the calculation. As far as, the XMS-CASPT2 correction is concerned, this would not be successful in accounting for the necessary dynamic electron correlation without a correct choice of the electronic wavefunction. Notice that the necessary benchmarking of the adopted CASPT2//CASSCF protocol has been performed in the past and documented in the literature.

This is now stressed in the main text.

Change 1. Change related to the methodological point above (main text, p4 line 76):

"Our MCQC models employ the well-established CASSCF zeroth-order wavefunction defined by the selection of a (12,12) active space including all the π -electrons and orbitals of the retinal chromophore. Although the trends in spectral properties discussed throughout the text are well reproduced at this level, we discuss the results obtained after multi-state (XMS-CASPT2) energy and geometrical corrections to the CASSCF geometries."

"...In general, there was not sufficient enthusiasm for the manuscript to be published in Nature Communications. I think that this study will be more suitable for a specialized journal but not for Nature Communications..."

Again, we respectfully disagree with the reviewer. The fact that it is possible, using a congruous set of QM/MM model, to reproduce the trend of the observed FQY, opens the route to the mechanistic understanding of the observed fluorescence enhancement along the series. We show that this be achieved by disentangling the factors determining the stability of an unprecedented and exotic reactive diradical intermediate (TIDIR) with respect to the fluorescent state (FS). It turns out that such stability is related to the specific electronic structure of TIDIR that, in contrast to FS, displays a charge localized on the chromophore Schiff base moiety.

These aspects are now stressed in the revised versions of the manuscript and supporting information material. Notice that, in connection with Reviewer #3 request, we have also performed our study at a different level of theory (SI-SA-REKS(2,2)) to show that the predicted stability trend is not dependent on the chosen quantum chemical treatment.

Reviewer #2 (Remarks to the Author):

Main points raised by the reviewer:

"...The authors present a computational investigation of fluorescence properties of Rhodopsins. The increase in fluorescence quantum yield is interpreted in terms of the features of the S1 potential energy surface, which are in turn related to the diabatic transition from charge transfer to diradical character when going from the Franck Condon point to the S1/S0 CI. The importance of the electrostatic effects of the amino acids close to the chromophore is clearly pointed out..."

We are grateful to the reviewer for his/her positive evaluation.

"... The authors should address the following remarks.

1) The authors are urged to give another careful read to the manuscript. In fact, there are several typos; several references and cross-links are missing in the supporting information; in the caption of figure 3, reference is made to a panel H which is not present in the figure, etc...."

We are grateful to the reviewer for pointing out these issues with the main manuscript and supporting information. The main manuscript and Supporting Information have now been thoroughly checked. The typo in the legend of Fig. 3 has been corrected.

"... 2) In general, there is no relationship between the diabatic coupling and the energy difference of two diabatic states. Then, the initial increase of the diabatic coupling between the Ag and Bu states when the alpha angles closes (and/or BLA increases) should not be related to the decrease in the Ag-Bu energy gap. Moreover, it is not clear why the Ag-Bu diabatic coupling should be modelled by a sinusoidal function (section S8), which is actually referred in the main text so as "a harmonic coupling function"..."

We thank the reviewer for his/her remark. We agree that there is no "a priori" relationship between diabatic coupling and energy difference between the diabatic states. However, such relationship is "imposed" by the postulated diabatic model (i.e. reaction mechanism) we propose. This model and the way in which it relates to the diabatic coupling is explained in the answer to point 3) below.

"... 3) According to Fig. 3C, it is postulated that the two diabatic states Ag and Bu crosses along the alpha coordinate going from the FS to the TIDIR point, and the S0 and S1 states exchange their characteristics. The authors should try to verify this assumption. For example, one may rely on the fact that, in this kind of crossing, the line integral of the nonadiabatic coupling vector should be equal to $\pi/2$. Alternatively, some kind of diabatization scheme could be employed..."

Indeed, in our scheme the diabatic states are associate with electronic characters or resonance formulas: a covalent/diradical (Ag, with the positive charge located in the Schiff base moiety) and a charge transfer (Bu, with the positive charge delocalized away from the Schiff base moiety) resonance formula. These are seen as components of a valence-bond-type wavefunction whose S_1 weights are inversely proportional to their stability. Following this definition, the diabatic energy changes are naturally defined as proportional to the weights of the resonance formulas and can be determined at each point along the computed path by following the variation in charge position/distribution. The fact that the weights invert in magnitude along the path indicates that

the diabatals must cross. Since the rotation about a C-C bond is periodic, it is also assumed, that the diabatals can be represented by sinusoidal functions.

More specifically, the pure Ag and Bu diabatals are assigned a charge on the iminium cation moiety of 1 (depicted as green color in Fig. 2E and 3C) and 0 (brown color), respectively. Therefore, we have found convenient to express the coupling such as it is maximized when the alpha angle reaches $45 + k \text{ Pi}$ degrees angle (with k and integer number), where the periodicity of the sinusoidal function arises from the pseudo-symmetry of the retinal chromophore model. From the comparison of Fig S7 (middle panels) and Fig S13 (right panels), it is easy to visualize the correspondence between the increase of the partial charge of the iminium cation in the S₁ state and the increasing green coloration in Fig S13, both associated with an increasing weight of the Ag diabatic state.

This is now better stressed in the main text.

Change 2. Change related to the methodological point above (main text, p7 line 154):

“We stress that this mechanistic model assumes no “a priori” relationship between diabatals energy difference and diabatic coupling but is a simple valence-bond type description of our adiabatic S₁ PES, assuming two pure resonance formulas (1A_g and 1B_u), whose weights are associated to the documented variation in positive charge distribution along the reaction path.”

“...4) Why the trajectories are started with zero kinetic energy? How it can be excluded that a realistic nonzero value of the starting kinetic energy could wash out the difference in the S₁ dynamics of Arch3 and Arch7?...”

We agree with the reviewer that the simulation of the S₁ dynamics must be based on a large set of different initial conditions. However, the aim of the present work is that of explaining the origin of the fluorescence in the Arch-set rather than predicting the excited state dynamics. As discussed in the literature (Manathunga et al., J. Chem. Theory Comput. 2016, 12, 2, 839-850. or Frutos et al. Proc. Natl. Acad. Sci. 2007, 104, 19, 7764-7769), single trajectories starting from the S₀ equilibrium structure with zero initial velocities (FC trajectories), is a valid tool when one needs to rapidly detect a barrierless or nearly barrierless isomerization coordinates. Accordingly, in our work FC trajectories are used to complement the static information provided by the S₁ torsional scans.

Change 3. Additional information has been included in the manuscript (main text, p7 line 166):

As discussed in the literature, these trajectories mimic the evolution of the center of the S₁ population and are useful to detect barrierless or nearly barrierless isomerization paths and validate the accuracy of computed torsional scans^{17,25}.

a new reference has also been added. This is **ref. 25**

Reviewer #3 (Remarks to the Author):

Main points raised by the reviewer:

"...The computationally applied methods can be considered of high-level, although nowadays almost standard within the community of computational quantum chemists interested in electronically excited states...."

"...the article lacks the required novelty and breakthrough type of result that is expected for a Nature Communications paper. Actually, if the authors continue developing a really new theoretical/computational tool (that was not the case of this manuscript) "for the in silico selection of highly fluorescent Arch3 variants", pointed out as future perspective in this work, then I would clearly suggest publication in Nat. Comm. ..."

No computational study reported to date has provided a clear path to the engineering of high fluorescence variants of Arch3 or other non-fluorescent wild type rhodopsins (some results were obtained via machine learning but, as such, they do not provide an atomistic mechanism). For this reason, we respectfully disagree with the reviewer about his/her conclusion that our manuscript lacks novelty. Such disagreement is detailed in point i below. On the other hand, as reported in point ii-iii, we have seriously considered the reviewer request and performed additional calculations using new/advanced methodologies that, however, reinforce the original conclusion.

(i) *The C13=C14 barriers are the result of a charge delocalization-then-confinement mechanism* never documented before. In other words, the demonstration that the excited state isomerization barrier is proportional to the fluorescence intensity "only" confirms that our models can be used to study Arch3-based optogenetic tools. Instead, it is the mechanism explaining the existence and magnitude of the barriers that provides the rational route to Arch3-mutants displaying enhanced fluorescence. Since, this central result has been overlooked, we have now stressed it in the revised version of the manuscript.

(ii) *A computational model for exploring the relationship between protein charge distribution and C13=C14 isomerization barrier is now presented.* To demonstrate the existence of a link between isomerization barrier and protein electrostatics we have developed an iterative optimization method capable of calculating the charge distribution inducing a specific energy difference between TIDIR and FS ($\Delta E_{\text{TIDIR-FS}}$). The model shows that, to increase $\Delta E_{\text{TIDIR-FS}}$, the cavity counterion charge must relocate *and* partially diffuse. This is what one observes when comparing the effects of Arch3 and Arch7 cavity residue charges in Fig. 4D, that clearly show a "diffusion" of the TIDIR stabilizing effect.

(iii) *SI-SA-REKS(2,2) geometry optimizations support the correctness of the estimated barrier progression along the Arch-set.* The SI-SA-REKS(2,2) quantum chemical method is an ensemble-DFT method never used for studying rhodopsin fluorescence. Such method is expected to yield excited state equilibrium geometries of the same quality of CASPT2 at an affordable computational cost. Accordingly, we employed SI-SA-REKS(2,2) to recompute the FS and TIDIR equilibrium structure for the Arch-set while maintaining the ESPF electrostatic embedding typical of our QM/MM models. The results of these new calculations confirm the conclusions of the original manuscript.

The following related changes have now been inserted in the manuscript.

Change 4. Change related to point i (main text, p10 line 235):

"We propose that this relocation and "diffusion" of the negatively charged counterion (a virtual counterion), coupled with a delocalization-then-confinement mechanism of the positive charge of

the chromophore, explains the regular change in isomerization barriers. In brighter Arch-3 variants (e.g. Arch-5 and Arch-7) the counterion is increasingly distant and more diffuse from the Schiff base moiety. In these variants, at $\alpha=0^\circ$ the chromophore positive charge is delocalized, and its centroid is close to the counterion, leading to a stabilization of FS. As soon as α progresses and approaches a 90° twist, the confinement of the chromophore charge in the Schiff base moiety gradually increases, causing its centroid to drift away from the virtual counterion, inevitably determining a de-stabilization of TIDIR.

To support this conclusion, we developed a basic model that allows to compute (i.e. optimize) the protein charge distribution inducing a specific $\Delta E_{\text{TIDIR-FS}}$ value. This is done by allowing the relocation and fragmentation (i.e. diffusion) of the negative charge of the main counterion to other cavity residue positions, in the hope to mimic what observed in Fig. 4E. This can be achieved by defining a scalar function of the cavity residue charge vector (\mathbf{q}) which returns $\Delta E_{\text{TIDIR-FS}}$. Such $\Delta E_{\text{TIDIR-FS}}(\mathbf{q})$ function is differentiated numerically to study how $\Delta E_{\text{TIDIR-FS}}$ responds to \mathbf{q} . As detailed in section S11, the problem of determining \mathbf{q} can be formulated as a constrained optimization. In Figure 5, we show how the optimization modifies the charge distribution of Arch3 to reproduce the $\Delta E_{\text{TIDIR-FS}}$ value computed for Arch7 at the zeroth-order level. The resulting \mathbf{q} shows that, for Arch3 to reproduce Arch7 excited state properties, a relocation of ca. 50% of D222 negative charge to other cavity residues is necessary supporting the conclusion that a relocation and diffusion of the counterion is indeed a determinant of the TIDIR destabilization in the Arch-set.”

and the inclusion of a new figure (Fig. 5):

Figure 5. Predicted change of counterion charge distribution of Arch3 QM/MM model to yield the Arch7 $\Delta E_{\text{TIDIR-FS}}$ value. A. $\Delta E_{\text{TIDIR-FS}}$ (SA2-CASSCF/AMBER level) change along the optimization steps leading from the Arch3 to the Arch7 value. The optimization is driven by the target $\Delta E_{\text{TIDIR-FS}}$ values ($\Delta E_{\text{TIDIR-FS}}^*$, indicated by a thin horizontal line) and corresponds to the minimization of the square of the scalar function $\Delta E_{\text{TIDIR-FS}}(\mathbf{q}) - \Delta E_{\text{TIDIR-FS}}^*$ (i.e. $\Delta \Delta E_{\text{TIDIR-FS}}(\mathbf{q})$) as a function of the cavity residue charge vector \mathbf{q} . Each iteration of the algorithm (two full iterations are reported in the panel) is divided in two parts; (i) first \mathbf{q} is optimized at fixed FS and TIDIR geometries via conjugated-gradient optimization such that $\Delta E_{\text{TIDIR-FS}}(\mathbf{q}) = \Delta E_{\text{TIDIR-FS}}^*$ and then (ii) the FS and TIDIR QM/MM model geometries are relaxed, and $\Delta E_{\text{TIDIR-FS}}$ is recomputed. If the difference between the $\Delta E_{\text{TIDIR-FS}}$ calculated after part ii and $\Delta E_{\text{TIDIR-FS}}^*$ is above a selected threshold, i-ii are repeated. The circles indicate the steps of part i, and the crosses indicate the results of geometrical relaxation of part ii. B. Final charge distribution obtained after convergence of the optimization above. Two-dimensional representation of the fraction of negative charges residing in the cavity residues is proportional to the radius of the blue circles (the main counterion D222 final charge is

indicated). The barycenter of the corresponding negative charge is shown as a red circle. The original localized Arch3 negative charge is indicated by the dotted open circle centered at the original counterion position.

Change 6. Change related to point ii (supplementary text, Section 11)

S11. Optimization of the cavity electrostatics

Starting from the QM/MM model of Arch3, we designed an optimization allowing the negative charge hosted by the main counterion (in this case the residue D222) to relocate and distribute on the other cavity residues to produce a specific value of $\Delta E_{\text{TIDIR-FS}}$. Briefly, we start from the QM/MM structures of the FS and TIDIR of Arch3, select a target $\Delta E_{\text{TIDIR-FS}}^*$ value for the optimization and optimize the electrostatics of the cavity such that the difference between the absolute values of $\Delta E_{\text{TIDIR-FS}}(\mathbf{q})$ and $\Delta E_{\text{TIDIR-FS}}^*$ is minimized. To achieve this goal, the total charges of the cavity residue of the Arch3 QM/MM model are represented by their MM force field charges per residue (q_i), which is 0 for neutral residues and +1 or -1 for charged residues. Conveniently, in Arch3 QM/MM model, the determined cavity (see Section S2), does not include positively charge residues, such that we can represent the model (virtual) counterion charge distribution by a vector ($\mathbf{q} = q_0, q_1, \dots, q_N$, where N is the number of residues), whose elements can host a negative charge comprised between 0 and -1. Furthermore, we impose that the global charge of the cavity (\mathbf{q}) must always equal -1 (as in the starting QM/MM model), to make sure that the final optimized model is as realistic as possible. Given this set of rules, the problem of finding a $\Delta E_{\text{TIDIR-FS}}^*$ value, can be formulated as a constrained optimization problem as follows:

$$\begin{aligned} & \min ((\Delta E_{\text{TIDIR-FS}}(\mathbf{q}) - \Delta E_{\text{TIDIR-FS}}^*)^2) & (16) \\ & \text{subject to } \sum_i^N q_i = -1; q_1, q_2, \dots, q_N \leq 0 \end{aligned}$$

When a residue hosts a fraction of negative charge different from 0 such fraction is equally distributed amongst the atom of the residues. Since $\Delta E_{\text{TIDIR-FS}}$ is calculated at the SA2-CASSCF/AMBER level and no gradient of the CASSCF wavefunction is available with respect to the charges, the minimization is performed by computing the gradient numerically, with the following two-point formula:

$$\frac{(\Delta E_{\text{TIDIR-FS}}(\mathbf{q}) - \Delta E_{\text{TIDIR-FS}}^*)^2 - (\Delta E_{\text{TIDIR-FS}}(\mathbf{q} + \Delta \mathbf{q}) - \Delta E_{\text{TIDIR-FS}}^*)^2}{\Delta \mathbf{q}} \quad (17)$$

where $\Delta \mathbf{q}$ is set to 0.001. The optimization is carried out at fixed geometry and all the $\Delta E_{\text{TIDIR-FS}}(\mathbf{q})$ are therefore evaluated via single point calculations. Since $\Delta E_{\text{TIDIR-FS}}$ is a reaction energy, for all tested (\mathbf{q}) two single point calculations needs to be performed, at FS and TIDIR geometries. The optimization of the charge distribution is performed using python Scipy³⁶ code interfaced with MOLCAS, which performs QM/MM calculation and computes the gradient which is then used by the trust region algorithm³⁷ implemented in Scipy to perform the minimization and find optimal (\mathbf{q}) according to Eq. 16.

In the main text we discuss the application of this procedure to target the $\Delta E_{\text{TIDIR-FS}}$ of the brightest mutant Arch7.

The algorithm presented is composed of two parts. Part i consists of the (\mathbf{q}) optimization procedure discussed above. Along the optimization, only (\mathbf{q}) is relaxed and the geometries are kept fixed. In part ii, once a solution (\mathbf{q}) has been found, the QM/MM geometries of FS and TIDIR in the new electric field of the charges (\mathbf{q}) are relaxed on the S_1 PES. At this points, $\Delta E_{\text{TIDIR-FS}}$ is recomputed. At the end of parts i and ii, if $\Delta E_{\text{TIDIR-FS}}$ is significantly different from $\Delta E^*_{\text{TIDIR-FS}}$, (we consider a tight energy difference threshold of $0.1 \text{ kcal mol}^{-1}$), i-ii are repeated until $\Delta E_{\text{TIDIR-FS}} = \Delta E^*_{\text{TIDIR-FS}}$ according to the selected threshold.

Change 7. Change related to point iii (supplementary text, Section 5)

To corroborate the validity of our BLA "correction vector", we re-optimized the geometries of the SA2-CASSCF/AMBER S_1 stationary points (FS and TIDIR) using the state-interaction state-averaged spin-restricted ensemble-referenced Kohn–Sham method (SI-SA-REKS)²⁴ to treat the QM moiety. The method has been benchmarked on a retinal chromophore model ground and excited reaction paths documenting an accuracy comparable to wavefunction-based multi-state multiconfigurational methods²⁵. Notice that, XMS-CASPT2 geometry optimizations at the QM/MM level are still unpractical and therefore we employ REKS for geometry optimization. In fact, the SI-SA2-REKS(2,2) approach that we employed (REKS/AMBER in short), two active electrons in two orbitals are used to describe the π - π^* excitation which is main determinant of the RPSB chromophore photochemistry²⁶. Such active space also accounts for the static electron correlation while the dynamic electron correlation is included by using an exchange-correlation functional. As shown by Martinez et al. in two studies of Channelrhodopsin-2²⁷ and Bacteriorhodopsin²⁸, these features make SI-SA-REKS a valuable tool able to describe conical intersections and excited state reaction paths of the RPSB chromophore with performances comparable to other wavefunction-based multi-state multireference methods²⁹. In **Figure S8** panel A, we show that although the REKS/AMBER calculated $\Delta E_{\text{TIDIR-FS}}$ on the Arch-set is overestimated of few kcal mol^{-1} with respect to the XMS-CASPT2/AMBER method, the error is systematic, resulting in a parallel trend across the Arch-set. Similar conclusions hold when the energies of the REKS/AMBER optimized geometry are corrected at the XMS-CASPT2//SA2-CASSCF(12,12)/ 6-31G*/AMBER level of theory. In **Figure S8** panel B it is demonstrated that in our set of QM/MM models the trend in $\Delta E_{\text{TIDIR-FS}}$, is invariant with respect to the methodology employed to optimize the FS and TIDIR geometries, as shown by the equally accurate linear relationship between reaction energy and FQY.

and the inclusion of a new figure (Fig. S8):

Figure S8. SI-SA2-REKS(2,2) calculations. A. Dependency of the S_1 reaction energy ($\Delta E_{\text{TIDIR-FS}}$) on the level of theory of the QM/MM calculation. The blue curve is obtained from geometric interpolations (see Supplementary text S5) at the SA3-CASSCF(12,12)/XMS-CASPT2/ANO-L-vDZP/AMBER level of theory, the yellow curve from excited state geometry optimization at the SI-SA-REKS(2,2)/6-31G*/AMBER level and finally the red curve from energy correction at the SA2-CASSCF(12,12)/XMS-CASPT2/6-31G*/AMBER level of theory of the SI-SA-REKS(2,2)/6-31G*/AMBER optimized geometries. B. $\Delta E_{\text{TIDIR-FS}}$ calculated with different strategies holds the same linear relationship with experimental FQY.

"... the attempt to propose a highly novel excited state mechanism ends up being a rather usual (for rhodopsin systems) S_1 potential energy surface where (evidently) the presence of more or less stable (in energy) minimum can be related to the fluorescence quantum yield. This is not a surprise....It is especially surprising that some of the authors already published in 2012 about the origin of fluorescence in a rhodopsin model, by using almost the same techniques and level of theory, but did not mention it in the manuscript: Laricheva et al., J. Chem. Theory Comput. 2012, 8, 8, 2559–2563..."

As already discussed above, we have clarified that it is the molecular and electronic level mechanism allowing the regular increase of the barrier that represents the real novelty in our work. On the other hand, we need to stress that this is a first study where an entire series of engineered protein with experimentally observed and regularly increasing fluorescence has been systematically modelled/investigated using a congruous modeling protocol. While there have been previous QM/MM studies pointing to a fluorescent intensity generated by blocking an isomerization channel, these studies have never regarded an entire set of optogenetics tool that are already employed in the lab.

A second novelty regards the electronic character variation along the isomerization path. In fact, while Laricheva et al. discuss the presence of two minima on the S_1 PES, the electronic structure of such minima is represented by different levels of charge transfer. In contrast, the two minima located in the Arch-set demonstrate a different electronic structure variability in rhodopsins not

explicitly documented in previous studies (we refer to the purely diradical character of the TIDIR minimum located near the Coln). We therefore believe that our manuscript reports on fundamentally new knowledge that may have been overlooked. More specifically, since the spectroscopic states of rhodopsins is, in general, a charge transfer state, it is believed that the molecule will maintain and enhance the charge transfer character along the isomerization path. In our study we show that this not the case in the Arch-set and, according, to our conclusions in any rhodopsin that can be potentially fluorescent.

Change 8. This has been clarified in the manuscript (**main text, p4 line 89**):

"As it will be explained below, this previously unreported intermediate differs, in terms of electronic structure and topography, from the locally excited (LE) identified in a ring-locked derivative of bovine rhodopsin by Laricheva et al. ¹⁹ ..."

a new reference has also been added. This is **ref. 19**

"...Hence, I do not see any big advance in the field. Also, the diradical character of the highly twisted S₁ minimum structure raises up the curiosity whether a relatively high spin-orbit coupling value could lead to triplet population, hence making possible phosphorescence, apart from fluorescence. This aspect is not considered, and would actually increase eventually the novelty of the study, at least in the mechanistic point of view...."

We thank the reviewer for pointing out the possibility of an intersystem crossing (ISC) to a triplet state (T₁). We agree that this topic should be mentioned in the manuscript. The reason why ISC is not assumed to be a viable competitive process for S₁ emission is three-fold: (i) the substrate is fully organic without heavy atoms, hence, the spin-orbit coupling is small, (ii) T₁ and S₁ are both π-π* states and (iii) the TIDIR state, where the intersystem crossing may occur, is located very close to the S₁/S₀ conical intersections. In the conditions i-iii it is unlikely that the T₁ state gets populated as this would violate the El-Sayed rule. Furthermore, S₁ would not live long enough for the system to undergo ISC.

Change 9. This has been clarified in the manuscript (**main text, p3 line 60**):

"Despite the increased FQY of the investigated Arch variants, recent measurements of the excited state lifetime (ESL) of some of the variants were found to be in the time range of picoseconds. For this reason, we don't account in our calculation for T₁/S₁ intersystem crossing (ISC) as a viable competitive process to S₁ emission also considering that T₁ is a π-π* state with orbitals parallel (non-orthogonal) to those characterizing the S₁ state. Therefore, the singlet to triplet transition would be "forbidden" by the El-Sayed rule."

two new references have also been added. These are **ref. 10** and **ref. 11**

"...On the other hand, I do not feel in this case study the necessity to run a single semiclassical trajectory to show that photoisomerization is possible..."

We agree with the reviewer that the simulation of the S₁ wavepacket evolution, would require hundreds of semiclassical trajectories. However, the aim of the present work is that of explaining the origin of the fluorescence in the Arch-set rather than predicting the excited state lifetime or the isomerization quantum yield. As discussed in the literature (Manathunga et al., J. Chem. Theory Comput. 2016, 12, 2, 839-850. or Frutos et al. Proc. Natl. Acad. Sci. 2007, 104, 19, 7764-7769), single trajectories starting from the S₀ equilibrium structure with zero initial velocities (FC

trajectories), is a valid tool when one requires to rapidly detect the systems with barrierless or nearly barrierless isomerization coordinates.

Accordingly, in our work FC trajectories are used to complement the static information provided by the S_1 torsional scans.

Same as Change 2 above. Additional information has been included in the manuscript (**main text, p7 line 166**):

“As discussed in the literature, these trajectories mimic the evolution of the center of the S_1 population and are useful to detect barrierless or nearly barrierless isomerization paths and validate the accuracy of the energy profiles computed along the isomerization path.”

“...Also, more at a technical level, I did not fully get (also reading the supporting information) how S_1 minima and transition states were obtained. I would remind that such structures do require the calculation of the Hessian matrix, to finally check if all frequencies are positive (minimum) or if one single imaginary frequency is present (transition state). If the authors cannot clarify this issue, their structures are “only” approximated stationary points...”

We agree that this aspect is not clearly explained in the manuscript. Given the high computational cost of QM/MM Hessians (even at the 0th order SA2-CASSCF/MM level) we could not perform a standard Newton-Raphson search for a TS. However, in the revised version of the manuscript we do search for a TS at the less expensive *SI-SA-REKS(2,2) level starting from guess Hessians computed for the gas-phase chromophore. We obtain approximate transition structures located only few kcal mol⁻¹ above the TIDIR. After recomputing the energy barriers at the XMS-CASPT2/MM level we obtain values that are close to those estimated using the original SA2-CASSCF/MM calculations.*

Change 10. To clarify this issue, we included in the manuscript (**p4 line 94**):

“Due to the high computational cost of QM/MM analytical Hessians, the TS discussed throughout the text are approximated by the energy maxima along the relaxed scan connecting FS and TIDIR. These TSs must be considered approximate as it has not been possible to carry out a geometry optimization starting from a computed Hessian matrix as well as to compute a Hessian matrix at the end of the TS search.”

Finally, the authors corrected the CASSCF calculations by introducing the missing electronic dynamic correlation at CASPT2 level through the identification of a BLA “correction vector”. Wouldn’t that be simpler to run CASPT2/MM optimizations, nowadays accessible at relatively low computational cost by using e.g. the BAGEL software? In this way, they could also understand the feasibility of their CASSCF/MM approach, especially considering that few kcal/mol could completely change the picture in this case.

We thank the reviewer for discussing the possibility of using alternative approaches to introduce dynamic electron correlation in our models. Part of the response has already been addressed in **Change 7**.

Although the authors are aware of the possibility of using BAGEL, such software cannot presently be used in QM/MM calculations treating the electrostatic embedding via the ESPF operator used in our approach. On the other hand, geometry optimization at the CASPT2/MM level are not possible with our QM/MM models due to the unavailability of analytic nuclear gradients. For this

reason, we have adopted a BLA "correction vector" approach. To test the validity of such an approach, we have used the alternative strategy mentioned above that includes the ESPF electrostatic embedding. In fact, we used SI-SA-REKS to re-compute $\Delta E_{\text{TIDIR-FS}}$ and show that the results are not methodology-dependent. In SI-SA-REKS(2,2), the dynamic electron correlation is described by the exchange–correlation functional, while the static correlation is included by the configuration interaction between the closed-shell and open-shell singlet electronic configurations associated with 2 electrons, 2 orbital active space.

As shown in the new Figure S7 included in the Supporting Information, the most important quantity discussed in our research, $\Delta E_{\text{TIDIR-FS}}$, is affected quantitatively by the methodology employed for its calculation but in a systematic fashion. We found indeed that the trend in $\Delta E_{\text{TIDIR-FS}}$ is reproduced (i) at the XMS-CASPT2/ANO-L-vDZP//SA3-CASSCF(12,12)/6-31G* level of theory, from the BLA correction vector, (ii), after optimization at the SI-SA-REKS(2,2)/6-31G* level of theory, and (iii) after XMS-CASPT2//SA2-CASSCF(12, 12)/6-31G* energy correction of the REKS geometries.

Minor points:

-In the introduction, "four order of magnitude" should be modified as "four orders of magnitude".

We thank the reviewer for pointing this out. The sentence was revised accordingly.

-In Figure 1, the retinal chromophore is covalently linked to ϵ in panels A and B, but then it is linked to R in panel C. This is misleading. The specific protein residue should be instead considered.

We thank the reviewer for his/her suggestion.

Change 11. We modified Figure 1 to avoid a misleading interpretation as follows:

Figure 1. Photoisomerization mechanism of Archaeorhodopsins. A. Lewis formula representing the initial S_1 chromophore structure. B. Representation of the chromophore isomerization path. FS corresponds to the fluorescent state. TIDIR represents the photoisomerization channel located near CoIn. FS and TIDIR are represented by Lewis formulas displaying distinct degrees of double bond twisting and charge transfer. C. Main components of the reaction coordinate. BLA is numerically defined as the difference between the average single-bond length minus the average double-bond length along the C5 to N conjugated chain (for convenience, below we consider the BLA of the framed moiety exclusively). α is defined by the dihedral angle C12-C13-C14-C15.

-In Figure 2, the legend of panels A and B is insufficient to have an easy readability. Why are there two energy curves? Are they referred to S_1 and S_2 ?

We appreciate the reviewer remark and suggestion.

Change 12. We modified the legend of Figure 2, panels A and B, to clarify that the two energy curves shown refer to S_1 energies before and after multi-state correction. The modifications were included as follows:

Figure 2. Comparison between computed Arch3 and Arch7 S_1 isomerization paths. A. Variations in potential energy, charge distribution, free valence and oscillator strength along the Arch3 path. The energy profile in color is given after MS correction. B. Same data for Arch7. C. Main geometrical chromophore parameters for the FS fluorescent intermediates of Arch3 and Arch7 (values in square brackets). The S_1

positive charge fraction on the C-C-N moiety are also given. D. Same data for the photoisomerization channel TIDIR. E. Schematic "decomposition" of the Arch3 adiabatic energy profile of panel B in terms of diabatic states associated to Lewis formulas of the $\text{CH}_2=\text{NH}_2(+)$ minimal model. F. Schematic representation of the Coln region of Arch3 including the twisted diradical TIDIR along the relevant components of the reaction coordinate. The same coordinate also spans the Coln branching plane.

REVIEWERS' COMMENTS

Reviewer #1 (Remarks to the Author):

The authors have addressed some of my concerns. Although the manuscript was improved, the study itself is a typical black-box type poorly reasoned theoretical study. To that extent, the publication of this manuscript appears premature to me.

Still, there is no sufficient enthusiasm for the manuscript to be published in Nature Communications. I still think that this study will be more suitable for a specialized journal.

Reviewer #2 (Remarks to the Author):

I am quite satisfied with the corrections, additions and explanations provided by the authors.

Reviewer #3 (Remarks to the Author):

The authors performed a very complete and exhausting revision of their manuscript, not only writing better descriptions of some concepts and computational details, but also (and most importantly) taking care of running new calculations by which I can definitely understand their technical and scientific points of view.

Also, they made relevant efforts in convincing the reviewers about the novelty of the work (apparently, I was not the only one doubting).

Personally, I still doubt, more in general, about such topic being really helpful for increasing "Nature Communications" visibility. I mean that, in my point of view, this topic and even more the innerly specific technicalities highly necessary to explain the theoretical setup, computational efforts and

finally analytic tools, are not of easy understanding for a general scientific public, as it should be for "Nature Communications" standards.

Hence, I am still convinced that such (now) justified results are better suited for a more specific journal, maybe also within the Nature Publishing Group, where actually the interested theoretical reader could better understand all details, and thus reference the work more properly.

With the aforementioned personal doubts, I do not want to diminish the work and efforts of the authors. I just feel that this can be a really promising initial work on a novel topic that, inevitably, cannot be fully explained in a journal with a general scope, an expected large and somehow non-specific audience, and moreover with text size limitations, since it is a communication.

On the other hand, based on this more technical work, a future investigation/analysis of a wider amount of experimental data could definitely be written in a less technical way, hence being this time highly suitable for "Nature Communications", since it could also include some prediction that, by a matter of fact, is what experimentalists really expect from theoreticians in terms of useful results.

Point-by-point reply to the reviewers and list of changes

Reviewer #1 (Remarks to the Author):

Main points raised by the reviewer:

“The authors have addressed some of my concerns. Although the manuscript was improved, the study itself is a typical black-box type poorly reasoned theoretical study. To that extent, the publication of this manuscript appears premature to me. Still, there is no sufficient enthusiasm for the manuscript to be published in Nature Communications. I still think that this study will be more suitable for a specialized journal...”

Again, we respectfully disagree with the reviewer. There are two aspects that the reviewer disregards. The first is that, canonically, while methodologies such as HF, MP2 and, basically, all DFT methods are indeed black box, the methodologies used in our paper aren't. These are based on a multistate multiconfigurational wavefunction methods that are not black box. Also, the applied dynamic electron correlation correction is another, totally original, non-black-box part of the adopted methods. The second regards the fact that a completely novel excited state intermediate (TIDIR) has been located and proved to be critical for explaining the fluorescent quantum yield progression observed experimentally. Together with the present experimental interest in the systems investigated in our research (see also reply to reviewer #3) these two points support publication in Nat. Comm.

Reviewer #2 (Remarks to the Author):

Main points raised by the reviewer:

“I am quite satisfied with the corrections, additions and explanations provided by the authors.”

We are grateful to the reviewer for his/her positive evaluation.

Reviewer #3 (Remarks to the Author):

Main points raised by the reviewer:

“The authors performed a very complete and exhausting revision of their manuscript, not only writing better descriptions of some concepts and computational details, but also (and most importantly) taking care of running new calculations by which I can definitely understand their technical and scientific points of view.

Also, they made relevant efforts in convincing the reviewers about the novelty of the work (apparently, I was not the only one doubting).

Personally, I still doubt, more in general, about such topic being really helpful for increasing "Nature Communications" visibility. I mean that, in my point of view, this topic and even more the innerly specific technicalities highly necessary to explain the theoretical setup, computational efforts and finally analytic tools, are not of easy understanding for a general scientific public, as it should be for "Nature Communications" standards.

Hence, I am still convinced that such (now) justified results are better suited for a more specific journal, maybe also within the Nature Publishing Group, where actually the interested theoretical reader could better understand all details, and thus reference the work more properly.

With the aforementioned personal doubts, I do not want to diminish the work and efforts of the authors. I just feel that this can be a really promising initial work on a novel topic that, inevitably, cannot be fully explained in a journal with a general scope, an expected large and somehow non-specific audience, and moreover with text size limitations, since it is a communication.

On the other hand, based on this more technical work, a future investigation/analysis of a wider amount of experimental data could definitely be written in a less technical way, hence being this time highly suitable for "Nature Communications", since it could also include some prediction that, by a matter of fact, is what experimentalists really expect from theoreticians in terms of useful results."

We strongly believe that our research is of interest for the wide and interdisciplinary scientific community working in brain research and, more specifically, in Optogenetics. In fact, our results provide unprecedented mechanistic information on the modulation of fluorescence intensity in archaeal rhodopsins; the most used genetic encoded voltage indicators. Such wide interest is clearly supported by recent literature appeared in Nat. Comm. (e.g. see the just published article by Silapetere et al., *Nat Comm*, 2022, 13, 1, 1-20) that reports on different attempts to characterize the mechanism of such process. Accordingly, we think that Nat. Comm. is appropriate for reporting on our work.

In order to further stress the point above, we have included the following period and a new reference in the main text.

Change 1. Additional information has been included in the manuscript (main text, p 2, lines 53-54):

"...Recently, Hegemann and coworkers have investigated new Archon1 variants to elucidate the mechanism of fluorescence voltage sensitivity¹⁵..."

Change 2. Additional information has been included in the manuscript (main text, p 17, lines 370-371):

"...by experimentally investigating the fluorescence and voltage sensitivity mechanisms¹⁵..."

Change 3. References:

15. Silapetere, A. et al. QuasAr Odyssey: the origin of fluorescence and its voltage sensitivity in microbial rhodopsins. *Nat Commun* **13**, 1–20 (2022).